# Characterization and Evaluation of Taihe Black-Boned Silky Fowl Eggs Based on Physical Properties, Nutritive Values, and Flavor Profiles

**DOI:** 10.3390/foods13203308

**Published:** 2024-10-18

**Authors:** Libo Zhang, Mengru Xu, Fang Liu, Ru Li, Mahmoud M. Azzam, Xinyang Dong

**Affiliations:** 1Key Laboratory for Molecular Animal Nutrition of Ministry of Education, College of Animal Sciences, Zhejiang University (Zijingang Campus), Hangzhou 310058, China; zlb21131130@163.com (L.Z.); 18736551610@163.com (M.X.); liuf1210@zju.edu.cn (F.L.); 22217063@zju.edu.cn (R.L.); 2Animal Production Department, College of Food and Agriculture Sciences, King Saud University, Riyadh 11451, Saudi Arabia; mazzam@ksu.edu.sa

**Keywords:** egg quality, nutritional evaluation, amino acid, fatty acid, mineral, volatile flavor

## Abstract

Taihe black-boned silky fowl (TS) is a native chicken breed in China with more than 2000 years of history. The present study aimed to characterize and evaluate the physical, nutritional, and flavor properties of TS eggs with a comparison to two other commercial breeds. Eggs from TS (*n* = 60) crossbred black-boned silky fowl (CB, *n* = 60) and Hy-line Brown (HL, *n* = 60) were used for physicochemical analysis. The evaluation system was divided into four parts based on nutrient and flavor profiles: protein and amino acids, lipids and fatty acids, mineral elements, and flavor-related amino acids and volatile compounds. Results showed that TS eggs were typically associated with the lowest egg weight and the highest yolk color, as compared with CB and HL eggs. No differences were found in crude protein, crude fat, triglycerides, and cholesterol content between eggs from the different breeds, but these eggs were distinct in terms of the amino acid, fatty acid, and volatile flavor compound profiles. Moreover, the differences in amino acid and fatty acid profiles might contribute to the specific flavor of TS eggs. Evaluation results indicated that TS egg whites may be suitable as a protein source for premature infants and young children under three years old and TS egg yolks could be considered a beneficial dietary lipid source due to their potential anti-cardiovascular properties. Additionally, TS whole eggs could serve as a valuable source of selenium (Se), molybdenum (Mo), zinc (Zn), and phosphorus (P) for adults aged 18 to 65. However, TS and CB eggs showed inferior Haugh units, eggshell quality, and essential amino acid compositions for older children, adolescents, and adults. These findings provide a better insight into the health benefits of TS eggs and contribute to the breeding and nutrition regulation of TS breeds.

## 1. Introduction

Eggs are a vital part of the human diet due to their rich nutrient content and status as a primary protein source [1]. Research indicated that the physical and chemical characteristics of eggs vary significantly among different breeds, greatly influencing consumer preferences [2]. In China, consumers prefer indigenous chicken eggs over commercial ones [3]. A notable example is the Taihe black-boned silky fowl (TS), a rare local breed with a history exceeding 2000 years. TS is renowned for its nutritional value and pharmacological properties and is considered a miracle in traditional Chinese medicine for treating various ailments [4]. Recent studies have shown that indigenous chicken breeds often produce smaller eggs with higher nutritional value than commercial hens [5]. However, little research has compared TS eggs with those from commercial breeds. Despite the long history and recognized curative properties of TS, the health benefits of TS eggs remain unclear. Therefore, it is necessary to assess the quality of TS eggs.

Due to the limited egg production and brief peak laying period of TS, their eggs are sold at nearly twice the price of other chicken eggs [6,7]. However, the higher prices make these eggs more susceptible to fraudulent practices in the marketplace, negatively impacting egg quality, fair competition, and consumer preferences. In recent years, crossbred black-boned silky fowl (CB; Taihe black-boned silky fowl × recessive white chicken) have emerged as the main alternative to TS due to their similar appearance, higher egg production, and superior reproductive performance. Consumer preference for eggs from indigenous breeds supports the continued use and conservation of local breeds [8]. Therefore, distinguishing TS eggs from those of CB is crucial for protecting consumers’ rights and interests, as well as for preserving local genetic resources. However, there have been few attempts to characterize the physical properties and chemical composition of these two types of black-boned silky fowl eggs. To our knowledge, only one study has compared the quality of eggs from TS and CB layers [7]. Notably, this study focused on a limited range of nutrients, primarily minerals and lipids, and did not examine the comprehensive nutrient profile or the physical and flavor properties of the eggs.

In this study, eggs from TS were characterized and evaluated for their physical and chemical properties, including weight, Haugh units, yolk color, eggshell quality (encompassing thickness and strength), protein and amino acid content, lipid and fatty acid profiles, fat-soluble vitamin content, mineral content, and flavor profiles. These characteristics were compared with eggs from two other commercial breeds: Hy-line Brown (HL) and CB. The results enhance our understanding of the physical, nutritional, and flavor properties of TS eggs and may offer insights into their health benefits. Furthermore, this information can help differentiate TS eggs from those of commercial crossbred laying hens, thus preventing adulteration and ensuring fair competition in the marketplace.

## 2. Materials and Methods

### 2.1. Experimental Egg Sampling

A total of 180 eggs (60 for each breed) from TS, CB, and HL were kindly supplied by Xichang Fengxiang Poultry Co., Ltd. (Taihe, China); Wens Foodstuff Group Co., Ltd. (Xinxing, China); and Nuofeng Agricultural Technology Co., Ltd. (Hangzhou, China), respectively. All the laying hens were 30–35 weeks old and fed a standardized corn- and soybean-based diet to minimize dietary variations. The hens were raised under similar environmental conditions, including temperature-controlled houses with a 16 h light/8 h dark cycle. All the laying hens were fed a corn- and soybean-based diet. All eggs were collected randomly from hens during their peak egg-laying period and stored at 4 °C until further use. Eggs were selected based on normal shape, intact shell, and absence of blood spots or other visible defects.

### 2.2. Physical Analysis

Physical analyses (weight, albumen height, Haugh units, yolk color, yolk proportion, eggshell thickness, and strength) were performed on all eggs individually (*n* = 60). The eggs were weighed and cracked, and a digital egg tester (DET-6000, NABEL, Kyoto, Japan) was used to measure albumen height, Haugh units, yolk color, and eggshell strength. Yolk color was measured using the Roche Yolk Color Fan, which provides a standardized scale from 1 (pale yellow) to 15 (dark orange). Eggshell strength was determined by measuring the force required to crack the shell, expressed in kilograms of force (kgf). The thickness of the eggshell, excluding the shell membrane, was measured at the midpoint of the egg using a digital micrometer with a precision of 0.01 mm.

### 2.3. Chemical Composition Analysis

Egg whites and yolks were separated manually and stored at −80 °C until further chemical analysis. For chemical analyses (proteins and amino acids, fats and lipids, fat-soluble vitamin content, fatty acids, mineral elements, and volatile compounds), three individual whole eggs (without eggshells), egg whites, or egg yolks from the same breed were pooled together and mixed to form a single sample. Each analysis had eight experimental units (pools) for statistical analyses (*n* = 8). The analytical determinations were repeated in triplicate for each sample in the technique.

#### 2.3.1. Proteins and Amino Acids Analysis

Crude protein content in whole egg and egg white samples was determined using the traditional Kjeldahl method, as described by the Association of Official Analytical Chemists (AOAC) [9]. Egg white samples were stored at −80 °C and slowly thawed at 4 °C. A 50 μL aliquot of each sample was added to 400 μL of a pre-cooled methanol–acetonitrile solution (1:1, *v*/*v*). Following this, 50 μL of a 50 μM 16-isotope internal standard mixed solution was added. The mixture was vortexed for 60 s and then incubated at −20 °C for 1 h to precipitate proteins. The samples were then centrifuged at 14,000 rcf and 4 °C for 20 min. The resulting supernatant was freeze-dried and stored at −80 °C. Subsequent analysis of amino acids was conducted by Shanghai Applied Protein Technology Co., Ltd. (Shanghai, China), using a liquid chromatography tandem mass spectrometry (LC-MS/MS) system in multiple reaction monitoring (MRM) mode.

#### 2.3.2. Fats and Lipoids Analysis

Crude fat content in egg yolk samples was determined using the Soxhlet extraction method [9]. Concentrations of triglycerides and cholesterol in egg yolk samples were measured spectrophotometrically (UV-2000, Unico Instruments Co., Ltd., Shanghai, China) using commercial diagnostic kits (Nanjing Jiancheng Bioengineering Institute, Nanjing, China). For lipid extraction, 100 mg of each egg yolk sample was transferred into a 5 mL centrifuge tube. Briefly, 1000 μL of a methyl tert-butyl ether (MTBE)–methanol–water mixed solution (10:2:5, *v*/*v*) was added. The samples were ultrasonicated in an ice water bath for 10 min, followed by 1 min of rapid freezing under liquid nitrogen. This process was repeated three times. Subsequently, the samples were ground for 1 h at −20 °C and then centrifuged at 13,000 rpm for 15 min. The supernatant was transferred to a new tube. Then, 100 μL of a precooled dichloromethane–methanol solution (1:1, *v*/*v*) was added, followed by ultrasonication in an ice water bath for 10 min. After another centrifugation at 13,000 rpm for 15 min (4 °C), the supernatant was collected for phospholipid analysis. Quantitative analysis of phosphatidylcholine (PC), phosphatidylethanolamine (PE), sphingomyelin (SM), and ceramide (Cer) was performed by Shanghai Applied Protein Technology Co., Ltd. (Shanghai, China) using a liquid chromatography-electrospray ionization-tandem mass spectrometry (LC-ESI-MS/MS) system in multiple reaction monitoring (MRM) mode.

#### 2.3.3. Fatty Acids Analysis

After slowly thawing the egg yolk samples at 4 °C, an appropriate amount of each sample was transferred into a 10 mL centrifuge tube for lipid extraction. Briefly, 5 mL of dichloromethane–methanol solution (2:1, *v*/*v*) was added, and the mixture was vortexed to mix well, followed by washing with 2 mL of deionized water. The lower layer of the solution was collected and dried under nitrogen. Next, 2 mL of *n*-hexane was added along with an internal standard, and the mixture was methylated for 30 min. After adding 2 mL of deionized water, 2000 μL of the supernatant was collected and dried under nitrogen. The residue was then redissolved in *n*-hexane for fatty acid analysis. Quantitative analysis of fatty acids was performed by Shanghai Applied Protein Technology Co., Ltd. (Shanghai, China), using a gas chromatography tandem mass spectrometry (GC-MS/MS) system. The analysis was conducted on an Agilent 7890B gas chromatograph coupled with an Agilent 7000D triple quadrupole mass spectrometer (Agilent Technologies, Santa Clara, CA, USA). Chromatographic separation was achieved using an HP-5MS capillary column (30 m × 0.25 mm × 0.25 μm). The GC oven temperature program was as follows: initial temperature of 60 °C held for 1 min, then increased to 300 °C at a rate of 10 °C/min and held at 300 °C for 5 min. The MS was operated in electron impact (EI) mode with an ionization energy of 70 eV. Data acquisition and analysis were performed using Agilent MassHunter Workstation software (version B.08.00). The system was run in selected ion monitoring (SIM) mode for quantitative analysis of fatty acids.

#### 2.3.4. Fat-Soluble Vitamins Analysis

The egg yolk samples were retrieved from −80 °C storage, and 100 μL of each sample was taken and mixed with 10 μL of internal standard (IS-mix). Then, 400 μL of a pre-cooled methanol/acetonitrile solution containing 0.3% formic acid (1:1, *v*/*v*) was added, along with black ceramic beads. The mixture was homogenized twice (each for 20 s), vortexed for 30 s, and incubated at 4 °C for 10 min to precipitate proteins. After centrifugation at 14,000 rcf at 4 °C for 10 min, 400 μL of the supernatant was taken and combined with 400 μL of pre-cooled deionized water, vortexed for 30 s, and then 800 μL of the supernatant was drawn and passed through an Ostro SPE plate. The waste was discarded, and the elution was performed with 400 μL of isopropanol. The filtrate was dried and stored at −80 °C until further use. Fat-soluble vitamins A (VA), E (VE), K (VK), 25-OH-VD2, and 25-OH-VD3 were determined by Shanghai Applied Protein Technology Co., Ltd. (Shanghai, China), using a liquid chromatography-electrospray ionization-tandem mass spectrometry (LC-ESI-MS/MS) system. The analysis was performed on a UHPLC system (1290 Infinity II, Agilent Technologies) coupled to a triple quadrupole mass spectrometer (6470 Triple Quad, Agilent Technologies). Chromatographic separation was achieved on a Waters ACQUITY UPLC BEH C18 column (100 mm × 2.1 mm, 1.7 μm) maintained at 40 °C. The mobile phase consisted of 0.1% formic acid in water (A) and 0.1% formic acid in acetonitrile (B), with a flow rate of 0.3 mL/min. The gradient elution program was as follows: 0–1 min, 10% B; 1–6 min, 10–95% B; 6–8 min, 95% B; 8–8.1 min, 95–10% B; 8.1–10 min, 10% B. The injection volume was 2 μL. The mass spectrometer was operated in positive ion mode with multiple reaction monitoring (MRM). Data acquisition and analysis were performed using Agilent MassHunter Workstation software (version B.08.00).

#### 2.3.5. Minerals Analysis

Microwave digestion of whole egg samples (excluding the eggshell) was performed according to the procedures described by Giannenas et al. [10]. The digested egg samples were diluted to a final volume of 200 μL with ultrapure water. Trace elements, including magnesium (Mg), calcium (Ca), manganese (Mn), iron (Fe), copper (Cu), zinc (Zn), selenium (Se), chromium (Cr), molybdenum (Mo), and lead (Pb), were determined using inductively coupled plasma mass spectrometry (ICP-MS) (Perkin Elmer Inc., Waltham, MA, USA). Total phosphorus (P) concentration was determined by the ammonium vanadate method according to the national standards GB 5009.87-2016 of China [11].

#### 2.3.6. Volatile Flavor Compounds Analysis

The extraction of volatile compounds from whole egg samples was performed using a Headspace-SPME method, following the procedures reported by Xiang et al. [12]. After extraction, volatile compounds were analyzed using a gas chromatograph coupled with a time-of-flight mass spectrometer (GC-TOF-MS) system. Briefly, the headspace volatiles were separated using an Agilent DB-WAX (30 m × 0.25 mm × 0.25 μm) capillary column. The carrier gas was helium, with a flow rate of 1.0 mL/min, and the injection volume was 1 μL. The injector temperature was set at 245 °C, and the ion source temperature at 220 °C. The oven temperature program was as follows: 40 °C (3 min), ramped from 40 °C to 105 °C at 6 °C/min, 105 °C to 180 °C at 4 °C/min, 180 °C to 245 °C at 10 °C/min, and held at 245 °C for 5 min. Detection was performed in full scan mode (*m*/*z* 35–450) at an ionization voltage of 70 eV, with a scan rate of 15 spectra/s. To avoid the effects of instrumental detection signal fluctuations, samples were analyzed in a random sequence. QC samples were inserted into the sample queue to monitor and assess the stability of the system and the reliability of the experimental data.

### 2.4. Nutritional and Flavor Evaluation

#### 2.4.1. Proteins and Amino Acids

The evaluation of protein and amino acids in this study consists of three parts: protein and amino acid content, the ideal protein model, and amino acid standard spectra. Protein quality in egg whites was assessed based on the ideal protein model recommended by the Food and Agriculture Organization (FAO) and the World Health Organization (WHO). According to this model, high-quality protein should have an essential amino acid to total amino acids ratio (EAA/TAA) of less than 40%, and an essential amino acid to nonessential amino acids ratio (EAA/NEAA) greater than 60% [13].

Amino acid pattern spectrum evaluation is an assessment method that compares the composition of various amino acids to standard spectra [14,15]. Using the FAO/WHO’s established reference amino acid patterns for older children, adolescents, and adults, the three indices described below were calculated to measure the quality of amino acids in egg white samples [16]. The calculation formula of the essential amino acid index (EAAI) is as follows:
(1)$$\mathrm{EAAI}=\sqrt[{n}]{{\mathrm{II}}_{i=1}^{n}\mathrm{RAAi}}$$


The formula represents the geometric mean of the ratio of EAA content in the protein to that in the FAO/WHO standard reference pattern, serving as an indicator for assessing the amino acid balance. A higher EAAI indicates a more balanced amino acid composition, leading to higher protein quality and efficiency. The ratio of an amino acid (RAA) equals the content of a specific EAA in the sample protein/the content of that amino acid in the FAO/WHO reference pattern; the closer the RAA value is to 1, the closer the EAA content is to the WHO/FAO recommended levels. The relative coefficient of an amino acid (RCAA) equals the RAA of a specific EAA/the average RAA of all EAAs, where RCAA values above or below 1 indicate deviations from the amino acid pattern, with greater dispersion of RCAA values suggesting a more significant negative impact of a specific EAA on the amino acid balance [17].

#### 2.4.2. Fatty Acids

The evaluation of fatty acids comprised two components: fatty acid profile assessment and nutritional evaluation indices. This study utilized the atherogenic index (AI) to analyze the nutritional composition of fatty acids in eggs from three different chicken breeds. The AI, first introduced by Higuchi et al., is a crucial indicator for predicting cardiovascular and cerebrovascular risk [18]. When the AI exceeds 4, the likelihood of cardiovascular diseases significantly increases [19]. The calculation formula is as follows:AI = [C12:0 + (4 × C14:0) + C16:0]/∑UFA(2)
where C12:0 represents lauric acid, C14:0 represents myristic acid, C16:0 represents palmitic acid, and ∑UFA represents total unsaturated fatty acids.

#### 2.4.3. Minerals

This experiment used the index of nutritional quality (INQ) method to evaluate the nutritional value of mineral elements, including eight trace elements (Mn, Fe, Cu, Zn, Se, Cr, and Mo) and three major elements (Ca, Mg, and P) in eggs from three breeds [20,21]. This method reflects whether a certain nutrient in a food can meet human needs when the food provides sufficient calories [22]. In this study, dietary reference intakes (DRI) for mineral elements and energy are based on the recommended nutrient intakes or adequate intakes for the 18–65 age group established by the Chinese Nutrition Society (CNS) [23]. The calculation formula is as follows:INQ = (C_m_/DRI_m_)/(C_e_/DRI_e_) (3)
where C_m_ represents the content of mineral elements in 100 g of food, DRI_m_ represents the Dietary Reference Intake for the mineral element, C_e_ represents the caloric content in 100 g of food, and DRI_e_ represents the Dietary Reference Intake for calories. When INQ < 1, it indicates that the mineral element content is below the recommended supply level, and prolonged consumption may lead to a deficiency of that element; when 1 < INQ < 2, it indicates that the mineral element content is at or above the recommended supply level, suggesting good nutritional quality; when INQ > 2, it indicates that the food can be a good source of a certain mineral element [24,25].

#### 2.4.4. Flavors

Many amino acids have distinct flavors. It is commonly believed that the taste active value (TAV) can quantitatively assess the contribution of amino acids to food flavor [26,27]. The TAV is calculated using the following equation:TAV= C_t_/T_t_
(4)
where C_t_ represents the absolute concentration of the taste substance, and T_t_ represents the threshold value of the taste substance. A higher TAV indicates a more significant contribution of taste-active components to the overall flavor of the food [28].

The relative odor activity value (ROAV) method was used to screen for key volatile flavor compounds [29]. The compound contributing the most to the sample’s flavor was defined as follows:ROAV_max_ = 100(5)

For other flavor compounds:ROAV_i_ = (C_i_/C_max_) × (T_max_/T_i_) × 100 (6)
where C_i_ and T_i_ represent the relative content (%) and corresponding sensory threshold (mg/kg) of each volatile compound, respectively; C_max_ and T_max_ represent the relative content (%) and corresponding sensory threshold (mg/kg) of the compound contributing the most to the overall flavor of the sample. An ROAV greater than 1 indicates that the compound influences the food’s flavor, with higher ROAV values corresponding to a greater impact of that compound on the food’s flavor. Odor thresholds for the volatile flavor compounds were taken from online databases, including Chemical Book (https://www.chemicalbook.com/ProductIndex.aspx, accessed on 5 March 2024.) and the flavor database of Shanghai Jiao Tong University (https://mffi.sjtu.edu.cn/database, accessed on 7 March 2024).

### 2.5. Statistical Analysis

Statistical analysis was performed using SPSS 24.0 (SPSS Inc., Chicago, IL, USA). The Shapiro–Wilk test was employed to assess the normality of all datasets. For normally distributed data, homoscedasticity was assessed using Levene’s test. For data that followed a normal distribution and showed homoscedasticity, results are presented as mean ± standard error, and one-way analysis of variance (ANOVA) followed by Tukey’s multiple comparison test was used. For non-normally distributed data or data that violated the assumption of homoscedasticity, including volatile flavor compounds, results are presented as the median and interquartile range (IQR). The Kruskal–Wallis test was applied for these datasets, followed by Dunn’s post hoc test with Bonferroni correction for pairwise comparisons. *p* < 0.05 indicates a significant difference between groups, and *p* < 0.01 indicates an extremely significant difference.

## 3. Results

### 3.1. Physical Properties

As shown in Table 1, there were extremely significant differences in the physical properties of the eggs among the three breeds (*p* < 0.01). The CB and HL breeds had heavier eggs (45.37 g and 57.43 g, respectively), higher albumen height (3.96 mm and 7.74 mm, respectively), and lighter yolk color compared to the TS breed (*p* < 0.01). The percentage of egg yolk was higher in TS and CB than in HL (*p* < 0.01). The Haugh units in HL eggs were 88.39, which was significantly higher than in the other two breeds (*p* < 0.01). The eggshells from HL were more rigid than those from TS and CB breeds (*p* < 0.01). The thickness of TS eggshells was equivalent to that of HL, and both were significantly higher than that of CB (*p* < 0.01). No differences were found in the Haugh units, shell strength, and percentage of egg yolk between TS and CB breeds.

According to the standards set by the United States Department of Agriculture (USDA), the Chinese Ministry of Commerce (SB/T 10638-2011), and the Standardization Administration of China (GB/T 39438-2020), fresh eggs are categorized based on their Haugh unit values [30,31]. Eggs with Haugh unit values above 72 are classified as “Level 1” (AA), those with values between 60 and 72 are classified as “Level 2” (A), and those with values below 60 are classified as “Level 3” (B). As shown in Figure 1, the proportions of AA- and A-grade eggs for TS and CB were 85.41% and 83.64%, respectively, which are significantly lower than the 98.33% for HL (*p* < 0.05).

### 3.2. Characterization and Evaluation of Proteins and Amino Acids

Figure 2 represents the crude protein content of whole eggs and egg whites from three breeds, and there was no significant difference (*p* > 0.05). The amino acid composition of egg whites was further analyzed (Table 2). Except for methionine, the contents of the other 19 amino acids in the egg whites of the three breeds were significantly different (*p* < 0.01). Compared to HL, both TS and CB exhibited significantly higher levels of isoleucine, cysteine, and glutamine in egg whites (*p* < 0.01). The leucine content in TS and CB was significantly lower than in HL (*p* < 0.01, Table 2). No significant differences were observed in these four amino acids between TS and CB. Additionally, for the remaining 15 amino acids with significant differences in content among the three breeds, the trend was TS > CB > HL (*p* < 0.01).

The nutritive evaluation indices for protein and amino acids are shown in Table 3. The RAA values of leucine, sulfur-containing amino acids (methionine and cysteine), isoleucine, lysine, and tryptophan in TS, CB, and HL egg whites were less than 1. The RCAA values for leucine and tryptophan were the lowest in TS and CB egg whites, whereas in HL egg whites, lysine and tryptophan had the lowest RCAA values. The EAAI values for TS, CB, and HL eggs were 49.64, 57.19, and 56.04, respectively. The EAA/TAA ratios for TS, CB, and HL eggs were 24.43, 33.12, and 43.77, respectively, while the EAA/NEAA ratios were 32.37, 51.66, and 78.20, respectively.

Among NEAA, histidine, arginine, tyrosine, and cysteine are considered essential for preterm infants and young children under three years old. As depicted in Figure 3, the total content of histidine, arginine, tyrosine, and cysteine in TS egg whites was significantly higher (*p* < 0.01) than that in CB and HL egg whites.

### 3.3. Characterization and Evaluation of Lipids and Fatty Acids

The egg yolks from the three breeds contained 334.87 to 340.69 mg/g of crude fat, 14.91 to 15.92 mg/g of triglycerides, and 7.14 to 8.34 mg/g of total cholesterol (Table 4), with no significant differences (*p* > 0.05) among the breeds. However, the levels of phospholipids (PC and PE), sphingolipids (Cer and SM), and fat-soluble vitamins (25-OH-VD3 and VE) varied significantly (*p* < 0.01) among the three breeds (Table 4). The concentrations of PC, SM, and Cer in TS egg yolks were significantly higher than those in CB and HL (*p* < 0.01), with no significant differences between CB and HL. The PE and 25-OH-VD3 content in TS and CB egg yolks were comparable, both being significantly higher than in HL (*p* < 0.01). Conversely, the VE content in TS and HL egg yolks did not differ significantly, and both were significantly lower than in CB (*p* < 0.01).

The fatty acid content in egg yolks from different breeds is presented in Appendix A, with a total of 35 fatty acids detected (Figure 4). These included 14 saturated fatty acids (SFA), eight monounsaturated fatty acids (MUFA), and 13 polyunsaturated fatty acids (ω-3 and ω-6 PUFA). The fatty acid composition of egg yolks from the three different chicken breeds showed significant differences, with 19 fatty acids exhibiting significant (*p* < 0.05) or highly significant (*p* < 0.01) differences in content. The heat map in Figure 4 was constructed using GraphPad Prism (version 9.5.1) software.

Table 5 showed no significant differences (*p* > 0.05) in the contents of total fat acids (∑FA), total SFA (∑SFA), total MUFA (∑MUFA), total PUFA (∑PUFA), and total ω-6 PUFA (∑ω-6 PUFA) among egg yolks from the three breeds. The composition pattern of egg yolk fatty acids was consistent across breeds, specifically ∑MUFA > ∑SFA > ∑PUFA. The ratio of ω3/ω6 in TS egg yolks was significantly higher (*p* < 0.01) than in CB and HL. Eicosapentaenoic acid (EPA, C20:5N3) content in TS egg yolks was significantly higher (*p* < 0.01) than in CB and HL, while the docosahexaenoic acid (DHA, C22:6N3) and α-linolenic acid (C18:3N3) contents were comparable to those in CB and significantly higher (*p* < 0.01) than in HL. The calculated AI values for egg yolks from TS, HL, and CB were 0.45, 0.43, and 0.45, respectively.

### 3.4. Characterization and Evaluation of Mineral Elements

Among the 11 detected mineral elements, except for P, Cr, and Pb, the remaining eight elements exhibited highly significant differences (*p* < 0.01) in content across the whole eggs of the three breeds (Table 6). Compared to TS and CB, HL had significantly higher levels of Mg and Mo (*p* < 0.01) and significantly lower levels of Ca, Cu, and Zn (*p* < 0.01). There were no significant differences in the levels of these elements between TS and CB. The content of Mn in the whole eggs of CB was significantly higher (*p* < 0.01) than that of TS and HL. The Fe content in the whole eggs of CB was comparable to that of HL, both being significantly higher (*p* < 0.01) than that of TS. The Se content in the whole eggs of TS was significantly higher (*p* < 0.01) than that of CB and HL. The Pb content in the eggs of all three breeds was below the national food safety standards (GB 2762-2022) [32].

The INQ values of each mineral element are presented in Figure 5. In the context of dietary nutrition, distinct variations in micronutrient requirements between male and female cohorts were evident. Therefore, the INQ values were delineated separately. For the female demographic: In TS eggs, the INQ for Mg, Mn, Fe, and Cr stood below 1, while Ca and Cu were between 1 and 2; P, Zn, Mo, and Se exceeded 2. In CB eggs, Mg, Mn, and Cr exhibited INQs below 1, while Ca, Fe, Cu, and Zn registered between 1 and 2, and P, Se, and Mo surpassed 2. Similarly, within HL eggs, Mg, Mn, and Cr portrayed INQs less than 1, while Ca, Fe, Cu, Zn, and Se manifested between 1 and 2, and P and Mo exceeded 2. For the male demographic: In TS eggs, INQs for Mg, Mn, Fe, and Cr were below 1, whereas Ca and Cu occupied the 1 to 2 range, and P, Zn, Mo, and Se surpassed 2. CB eggs showcased INQs below 1 for Mg, Mn, Fe, and Cr, with Ca, P, and Cu situated between 1 and 2, and Zn, Se, and Mo surpassing 2. Similarly, in HL eggs, INQs for Mg, Ca, Mn, Fe, and Cr fell below 1, while Cu, Zn, and Se oscillated between 1 and 2, and P and Mo exceeded 2.

### 3.5. Characterization and Evaluation of Flavor

Amino acids influencing food flavor are primarily classified into four taste categories: umami, sweet, bitter, and sour. This study found that the umami TAV/total TAV value in TS egg whites was comparable to that in CB but significantly lower than in HL (*p* < 0.01, Figure 6). Conversely, the sweet TAV/total TAV value in TS egg whites was significantly higher than in CB and HL (*p* < 0.01, Figure 6), while the sour and bitter TAV/total TAV values were significantly lower (*p* < 0.01, Figure 6).

In the egg samples from the three chicken breeds, a total of 88 volatile compounds across eight categories were identified, including seven acids, 13 alcohols, two aldehydes, 21 alkanes, 17 benzenoids, six esters, eight ketones, and 14 oxygen-, nitrogen-, and sulfur-containing heterocyclic compounds (Appendix A). Further analysis of the content of these volatile flavor compounds revealed that alcohols were the primary constituents, accounting for 32.12% to 49.84% of the total volatile compounds, with no significant differences between the breeds (Table 7). TS showed no significant difference in the relative contents of acids, alcohols, alkanes, and ketones compared to CB and HL. However, CB exhibited significantly lower levels of acids (*p* < 0.05) and significantly higher levels of alkanes (*p* < 0.05) compared to HL. The content of aldehydes and benzenoids in TS eggs was comparable to HL but significantly higher than in CB (*p* < 0.05). The content of esters in TS eggs was almost identical to HL but significantly higher than in CB (*p* < 0.05), while the content of heterocyclic compounds in TS eggs was similar to HL but significantly lower than in CB (*p* < 0.05). These results suggested that the differences in the composition of flavor compounds among the three breeds were primarily focused on four volatile compounds: aldehydes, benzenoids, esters, and heterocyclic compounds.

Out of the 88 detected volatile flavor compounds, 53 compounds exhibited significant (*p* < 0.05) or highly significant (*p* < 0.01) differences in relative abundance among the breeds (Figure 7). Among these, 44 compounds with known thresholds were used to calculate ROAV values, revealing 10 compounds with ROAV > 1 (Table 8).

## 4. Discussion

External egg quality, including egg weight and shell quality, along with internal traits such as yolk color, size, and albumen viscosity, is crucial for consumers when selecting and cooking eggs [33]. Indigenous eggs are typically sold individually, with consumers often valuing smaller-sized eggs as a distinguishing feature [33]. The data indicated that TS eggs had a significantly lower weight compared to CB and HL, aligning with consumer perceptions and purchasing habits regarding indigenous eggs. Yolk color, derived from pigments such as oxidized carotenoids in the hen’s feed, is not directly related to nutritional content [34]. Nonetheless, a well-colored yolk (golden yellow to orange) is more likely to meet consumers’ expectations for egg quality [35]. In this study, the yolk color of TS eggs was darker compared to the other two breeds, which may have resulted from the lower laying frequency of indigenous breeds, leading to a higher concentration of pigment deposition within each yolk. Similarly, Bekele found that indigenous chicken eggs had darker yolks and lower weights compared to commercial eggs [36]. Several studies have demonstrated that breed significantly influences yolk color, with indigenous chicken breeds exhibiting significantly darker yolk coloration compared to crossbred and commercial chickens [37,38]. Egg yolks store most of the nutritional components, such as lipids; therefore, a higher yolk proportion indicates more dry matter and greater overall nutritional value, making these eggs more favored. In this study, TS eggs had the highest yolk proportion, aligning better with consumer preferences. Haugh units are a standard for grading the quality of fresh eggs and are internationally recognized for assessing the egg’s internal albumen quality [39]. They are derived from the regression relationship between egg freshness, egg white height, and egg weight. Eggshell quality, including shell thickness and strength, is a crucial external indicator influencing overall egg quality and directly impacting breakage rates during transportation [40]. This study found that TS and CB eggs exhibited lower Haugh units and shell strength but had higher proportions of grade B eggs compared to HL eggs. It is worth noting that consumers often reject eggs with Haugh units below 60 (grade B). These results indicated that TS and CB eggs showed poor internal albumen and eggshell quality, and efforts should be made to improve these qualities to avoid economic losses in the marketplace. Another interesting finding was that although the eggshell strength of TS and HL differed significantly, their eggshell thicknesses were comparable. We speculated that differences in eggshell strength between TS and HL might have been related to differences in their eggshell microstructure. However, further studies are needed to verify this hypothesis and explore potential mechanisms.

In the current study, no significant difference was found in the crude protein content of eggs from the investigated breeds. However, the amino acid content in egg whites varied significantly. Contemporary nutritional theories emphasize that the nutritional value of food proteins depends not only on the amino acid content but also on whether the types, contents, and proportions of EAAs align with human dietary needs [41]. Consequently, the protein nutritional value of egg whites was further evaluated using the WHO/FAO’s established amino acid standard spectra and ideal protein model. We found that the RAA values for five EAAs (e.g., leucine, sulfur-containing amino acids, iso-leucine, lysine, and tryptophan) were below 1, indicating that the egg whites from all three breeds could not fully meet the EAA nutritional requirements for older children, adolescents, or adults. This finding underscores the importance of consuming whole eggs rather than just egg whites to ensure a complete amino acid profile [42]. In the egg white samples of TS and CB, tryptophan exhibited the lowest RCAA score, followed by leucine, indicating that tryptophan and leucine were the first and second limiting amino acids, respectively. For adults and older children, adequate tryptophan intake is crucial for serotonin production, which plays a role in mood regulation and sleep [43]. Leucine is particularly important for muscle protein synthesis and metabolic health [44]. In HL egg whites, tryptophan and lysine were the first and second limiting amino acids based on RCAA values. Lysine is essential for proper growth and plays a vital role in calcium absorption and the formation of collagen, which is particularly important for bone health in growing children and older adults [45]. The EAAI values for CB and HL egg whites were similar and higher than those for TS. HL egg whites had an EAA/TAA ratio of about 40% and an EAA/NEAA ratio of over 60%. These findings suggest that HL egg whites may provide a more balanced amino acid profile for adults and older children, potentially supporting overall protein synthesis and metabolic functions more effectively [46]. Interestingly, our results showed that histidine, arginine, tyrosine, and cysteine were more abundant in TS egg whites compared to those in CB and HL egg whites. These NEAAs are particularly important for premature infants and children under the age of three, who have higher demands for these amino acids due to limited internal synthesis capacity [47]. Histidine is crucial for growth and tissue repair, arginine supports the immune system and wound healing, tyrosine is a precursor for neurotransmitters, and cysteine is important for antioxidant production [48]. Therefore, TS egg whites may be more suitable as a complementary food for premature infants and young children, potentially supporting their specific nutritional needs during critical developmental stages. In summary, while HL egg whites exhibited a more balanced EAA composition closely matching the FAO/WHO ideal protein standard for adults and older children, TS egg whites showed a potentially advantageous NEAA profile for younger children and premature infants. These findings highlight the importance of considering age-specific nutritional needs when recommending egg consumption and underscore the potential for breed-specific eggs to meet diverse nutritional requirements across different age groups.

Lipids in food include fats (triglycerides) and lipoids (sterols, phospholipids, sphingolipids, fat-soluble vitamins, etc.). Phospholipids (e.g., PC and PE) and sphingolipids (e.g., SM and Cer) play vital roles in regulating blood lipids, reducing cholesterol, combating cancer, and preventing dementia. Among these, PC is hailed as the “third nutrient” alongside proteins and vitamins. Previous reports found that eggs from locally bred Italian breeds exhibit higher cholesterol levels compared to crossbred breeds [49]. In the present study, no significant differences were found in crude fat, triglycerides, or cholesterol content among the three breeds of eggs tested, which contrasts with previous reports. However, the contents of PC, SM, PE, and Cer in the yolk of TS were the highest, followed by the yolks of CB and HL, indicating that the TS egg yolks had stronger antiatherosclerosis and antithrombosis abilities. Fatty acids are essential components of phospholipids and sphingolipids. The ratio of ω-3 PUFAs to ω-6 PUFAs (ω-3/ω-6) is an important indicator of the nutritional value of fatty acids [50]. The CNS recommends a ratio of 1:4~6 [23]. Our results showed that the ω-3/ω-6 ratio in the egg yolks of all three investigated breeds was below the recommended value; however, TS egg yolks had a higher ratio compared to the other two breeds. C18:3N3 is an essential fatty acid for the human body, providing multiple physiological functions such as reducing blood lipids and preventing thrombosis [51]. DHA, commonly known as “brain gold”, and EPA, often called the “artery scavenger”, contribute to human health and disease prevention. They reduce fat accumulation, decrease cardiovascular disease risk, slow inflammatory responses, promote neural development, and combat cancer [52]. The TS egg yolks contained the highest levels of DHA, EPA, and C18:3N3, followed by CB and HL egg yolks. AI is a key indicator for predicting cardiovascular and cerebrovascular disease risk, incorporating factors related to coronary heart disease, with values over 4 significantly increasing the likelihood of these diseases [19]. In this study, the AI of eggs from all three breeds was less than 0.5, indicating that their yolk fatty acid composition aligns with human health requirements. Overall, the fatty acid composition of the egg yolks from all three breeds poses no threat to human health, with TS egg yolks being particularly favorable as a lipid source for human consumption. While our study provides a comprehensive analysis of the fatty acid profile, it is important to note that direct clinical evidence supporting the cardiovascular health benefits of TS eggs specifically is limited. However, several studies have investigated the effects of eggs with similar fatty acid profiles on cardiovascular health. A randomized controlled trial by Blesso et al. (2013) found that consumption of eggs enriched in ω-3 PUFA improved lipid profiles and reduced inflammation markers in individuals with metabolic syndrome [53]. Moreover, another study found that healthy individuals who consumed ω-3 PUFA-enriched eggs for three weeks experienced improvements in microvascular reactivity, blood pressure, and triglyceride levels, suggesting potential cardiovascular benefits [54]. While these studies support the potential health benefits of eggs with fatty acid profiles similar to TS eggs, further research is needed to directly assess the impact of TS eggs on cardiovascular health. Future clinical trials specifically examining the effects of TS egg consumption on cardiovascular risk factors would provide more definitive evidence for our claims. CB egg yolks are notable for their high content of 25-OH-VD3 and VE, making them a good source of dietary fat-soluble vitamins. The potential health implications of these higher vitamin levels also warrant further investigation through clinical studies.

Minerals are one of the seven essential nutrients for human health. Macro elements (P, Ca, Mg, K, and Na) are crucial for development and maintenance, while trace elements (Mn, Zn, and Cu) act as vital catalysts in biochemical processes and metabolic activities [55,56]. Since the body cannot synthesize minerals, they must be obtained from food, drugs, or other external sources [57]. Dietary intake is the primary means to meet mineral needs, making mineral-rich foods highly valuable [58]. The INQ score chart indicated that the nutritional quality of Mg, Mn, and Cr in all eggs failed to meet the dietary requirements for individuals aged 18 to 65. Additionally, TS eggs did not meet the Fe requirements for the overall population, while CB and HL eggs did not meet the Fe requirements and HL eggs did not meet the Ca requirements for the female subgroup. According to the INQ values, CB eggs were a good dietary source of Mo and Se, HL eggs were rich in Mo and P, and TS eggs provided beneficial amounts of Se, Mo, Zn, and P for the target population, with Se being particularly notable (INQ > 5). Based on the aforementioned data, TS eggs had a higher mineral nutritional value than CB and HL eggs.

Flavor, an essential food attribute, results from a blend of volatile and non-volatile components, significantly shaping consumer preferences [59,60]. According to the TAV values of flavor amino acids, TS eggs exhibited the lowest levels of umami, sour, and bitter tastes but excelled in sweetness compared to CB and HL eggs, indicating a superior overall flavor amino acid profile, consistent with the findings of Yang et al. [61]. This observation likely contributed to the consumer preference for indigenous chicken eggs over commercially high-yield alternatives. A total of 88 volatile flavor compounds were identified in the studied eggs, with 53 showing statistically significant changes across the three different breeds. However, the relative content of these volatile flavor compounds does not always reflect their impact on overall odor. The evaluation must also consider their odor threshold levels, as different compounds have varying thresholds. Some compounds, even at low concentrations, can significantly influence odor due to their low detection thresholds. The ROAV is a method established to determine the key volatile flavor compounds in food by combining the sensory perception of compounds [62]. This method has been increasingly applied in recent years to identify key volatile flavor compounds in various foods [63,64,65,66]. Compounds with a ROAV value greater than one influence food odor, with higher ROAV values indicating a larger impact [67]. Based on the ROAV values of identified volatile flavor compounds, we identified ten key volatiles: ethyl isovalerate, 1-octen-3-ol, ethyl 2-methylpropionate, 1-hexanol, 1-heptanol, 2-pentylfuran, styrene, 1,2-dichloropropane, 1,4-dichlorobenzene, and indole. Ethyl isovalerate, 1-octen-3-ol, ethyl 2-methylpropionate, and 1-hexanol were common in TS, CB, and HL eggs, imparting fruity, fishy, earthy, grassy, and herbal odors. Ethyl isovalerate contributed the most flavor to TS and HL eggs, providing apple, pineapple, and banana odors. In contrast, 1-octen-3-ol was the most significant contributor to CB eggs, adding fishy, earthy, and grassy odors. Compared to HL eggs, TS and CB eggs shared 1-heptanol and 2-pentylfuran, which added natural, nutty, and vegetable odors. Styrene was a unique key volatile for TS eggs, imparting balsamic and floral odors, while CB eggs were distinct for their 1,2-dichloropropane, 1,4-dichlorobenzene, and indole, which contributed sweet, aromatic, camphor, and fecal odors. Volatile flavor compounds were primarily produced through the metabolism of amino acids and fatty acids. For instance, 1-octen-3-ol, 1-hexanol, and 1-heptanol are derived from the oxidative degradation of fatty acids, while the Maillard reaction and Strecker degradation of amino acids are related to the formation of many heterocyclic compounds (e.g., 2-pentylfuran and indole) [68]. Previously, we described that the amino acid and fatty acid profiles of TS eggs were significantly different from those of CB and HL eggs. This difference could explain the variation in egg odors among the three breeds. To provide more insight into the potential mechanisms linking the unique flavor profile of TS eggs to their nutritional composition, we can elaborate on the following points. 1. Amino acid contribution: The higher levels of certain amino acids in TS eggs contribute to their distinct flavor profile. For example, glycine and alanine, which were found in higher concentrations in TS eggs, are known to impart sweet flavors [69]. This aligns with our observation of TS eggs excelling in sweetness compared to CB and HL eggs. The lower levels of glutamic acid in TS eggs may explain their reduced umami taste, as glutamic acid is a key contributor to umami flavor [70]. 2. Fatty acid influence: The fatty acid composition of TS eggs also plays a crucial role in their flavor profile. The higher proportion of PUFAs in TS eggs, particularly ω-3 PUFA, may contribute to the formation of certain volatile compounds. For instance, the breakdown of ω-3 PUFA can lead to the production of hexanal and other aldehydes, which contribute to the ‘grassy’ and ‘fresh’ notes in the egg flavor [71]. 3. Maillard reaction products: The unique combination of reducing sugars and amino acids in TS eggs can lead to specific Maillard reaction products during cooking. The presence of specific amino acids such as lysine and arginine, combined with reducing sugars, can form unique flavor compounds such as pyrazines and thiazoles, which contribute to the roasted and nutty flavors in cooked eggs [72]. 4. Sulfur-containing compounds: The levels of sulfur-containing amino acids in TS eggs may influence the formation of volatile sulfur compounds. Methionine and cysteine can break down to form compounds such as dimethyl sulfide and hydrogen sulfide, which, at low levels, contribute to the characteristic egg aroma [73]. 5. Breed-specific metabolic pathways: The genetic makeup of TS chickens may lead to unique metabolic pathways that influence egg composition. Certain enzymes or metabolic processes specific to TS chickens could result in the formation of unique flavor precursors or directly contribute to the formation of flavor compounds such as styrene, which was identified as a key volatile unique to TS eggs [74]. While these mechanisms provide insight into the link between TS eggs’ nutritional composition and their unique flavor profile, further research is needed to fully elucidate the complex interactions between nutrients and flavor compounds in eggs from different breeds. Future studies could employ metabolomics approaches to trace the formation of key flavor compounds from specific nutrient precursors in TS eggs.

This study offers several key strengths, including a comprehensive comparison of nutritional and flavor profiles of eggs from three distinct chicken breeds (TS, CB, and HL), employing advanced analytical techniques and internationally recognized standards and providing insights into potential mechanisms underlying consumer preferences for indigenous eggs. However, we acknowledge limitations such as the lack of direct assessment of health impacts or consumer acceptability, the focus on a limited number of breeds, the absence of established causal relationships between nutritional composition and flavor profiles, and the omission of potential variations due to factors such as hen age, feed composition, or environmental conditions. Future research directions could include conducting clinical trials to assess health impacts, performing sensory evaluations, investigating the effects of farming practices and environmental factors on egg characteristics, and exploring breed-specific egg products. While our study provides valuable insights into comparative egg characteristics from different chicken breeds, further research is needed to fully understand the implications for human nutrition and the poultry industry.

## 5. Conclusions

Overall, the present study demonstrated differences in the physical, nutritional, and flavor properties of TS, CB, and HL eggs. TS eggs were typically associated with the lowest egg weight and the highest yolk color compared to CB and HL eggs. No differences were found in crude protein, crude fat, triglycerides, and cholesterol content among eggs from the different breeds, but the eggs were distinct in terms of amino acids, fatty acids, and volatile flavor compound profiles. The differences in amino acid and fatty acid profiles may contribute to the specific flavor of TS eggs; however, the underlying mechanisms require further research. The evaluation results indicated that TS egg whites might be suitable as a protein source for premature infants and children under three years old. TS egg yolks could be considered a good dietary lipid source due to their potential to prevent cardiovascular disease, while TS whole eggs could be a good source of Se, Mo, Zn, and P for individuals aged 18 to 65. Additionally, our data revealed that TS and CB eggs showed inferior Haugh units, eggshell quality, and essential amino acid compositions for older children, adolescents, and adults. Given the increasing consumer preference for TS eggs and their importance in maintaining local genetic resources and biodiversity, future efforts in breeding and nutrition regulation of TS breeds are needed to improve these deficiencies.

## Figures and Tables

**Figure 1 foods-13-03308-f001:**
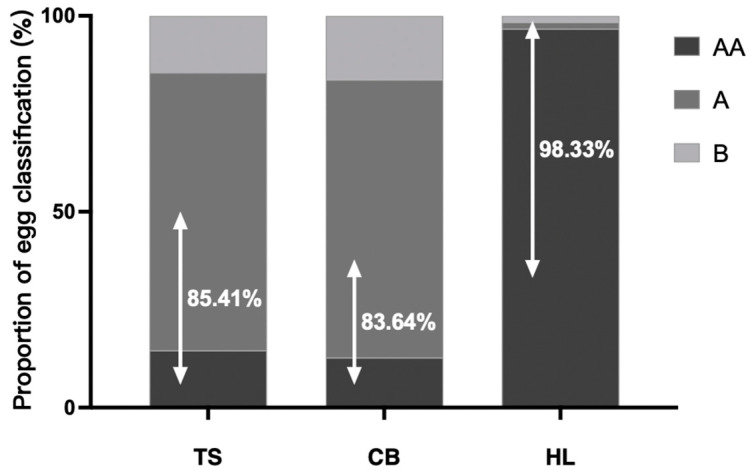
Grading of eggs based on Haugh unit values in different breeds. TS, Taihe black-boned silky fowl; CB, crossbred black-boned silky fowl; HL, Hy-line Brown. Black, AA (Haugh unit values above 72); dark gray (Haugh unit values between 60 and 72); light gray, B (Haugh unit values below 60). The percentages in the figure represent eggs rated as “AA” and “A” levels, accounting for the proportion of each variety’s egg samples. The proportions of AA- and A-grade eggs for TS and CB were significantly lower than for HL (*p* < 0.05).

**Figure 2 foods-13-03308-f002:**
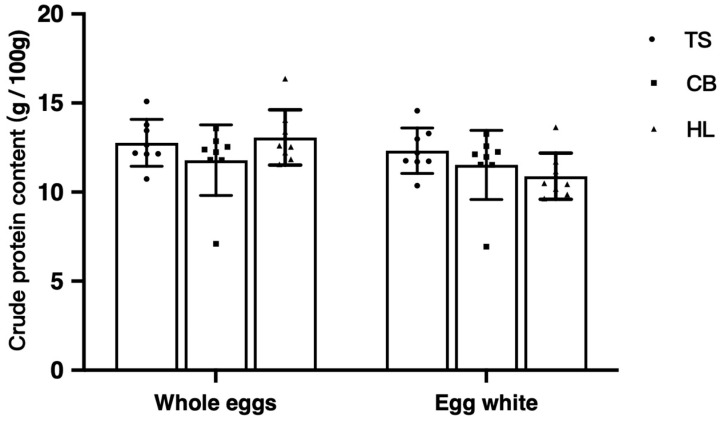
Crude protein content of Hy-line Brown, Taihe, and Crossbred black-boned silky fowl eggs. TS, Taihe black-boned silky fowl; CB, crossbred black-boned silky fowl; HL, Hy-line Brown. There was no significant difference (*p* > 0.05).

**Figure 3 foods-13-03308-f003:**
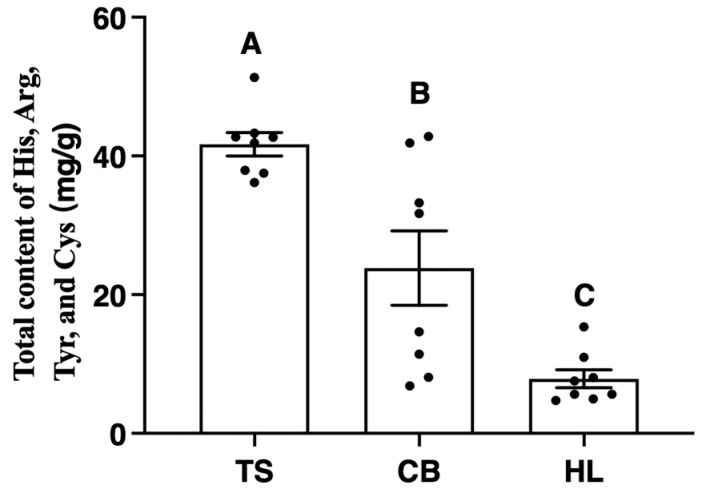
The total content of histidine, arginine, tyrosine, and cysteine in egg whites from different breeds (mg/g). TS, Taihe black-boned silky fowl; CB, crossbred black-boned silky fowl; HL, Hy-line Brown; His, histidine; Arg, arginine; Tyr, tyrosine; Cys, cysteine. A–C means significant at *p* < 0.01.

**Figure 4 foods-13-03308-f004:**
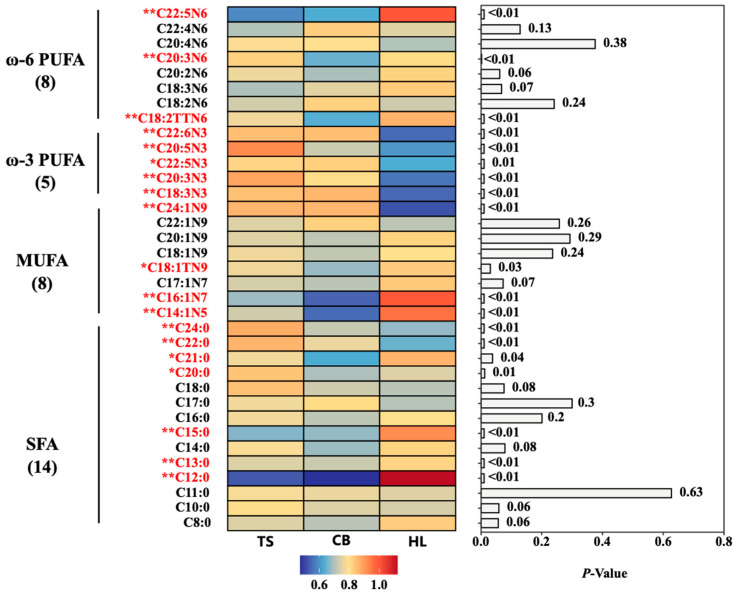
Fatty acid composition in egg yolks from Hy-line Brown, Taihe, and crossbred black-boned silky fowl. TS, Taihe black-boned silky fowl; CB, crossbred black-boned silky fowl; HL, Hy-line Brown; ω-3 PUFA, ω-3 polyunsaturated fatty acids; ω-6 PUFA, ω-6 polyunsaturated fatty acids; MUFA, monounsaturated fatty acids; SFA, saturated fatty acids. * Means significant differences (*p* < 0.05) within each fatty acid; ** Means significant at *p* < 0.01 within each fatty acid.

**Figure 5 foods-13-03308-f005:**
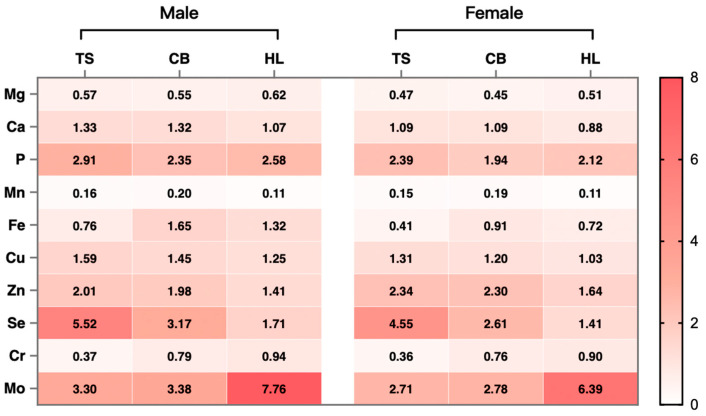
INQ values of mineral elements in whole eggs from different breeds. INQ, index of nutritional quality, calculated by the recommended nutrient intakes or adequate intakes for the 18–65 age group established by the Chinese Nutrition Society (CNS, 2023); TS, Taihe black-boned silky fowl; CB, crossbred black-boned silky fowl; HL, Hy-line Brown.

**Figure 6 foods-13-03308-f006:**
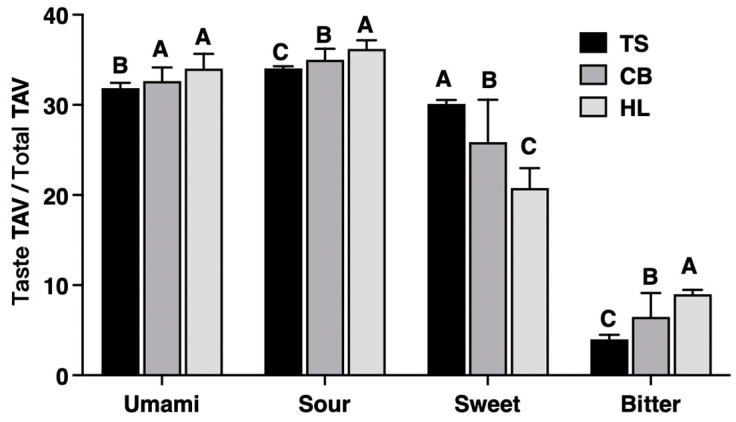
TAV difference in egg whites among three different breeds. TS, Taihe black-boned silky fowl; CB, crossbred black-boned silky fowl; HL, Hy-line Brown; TAV, taste active value. A–C Means significant at *p* < 0.01 within each taste.

**Figure 7 foods-13-03308-f007:**
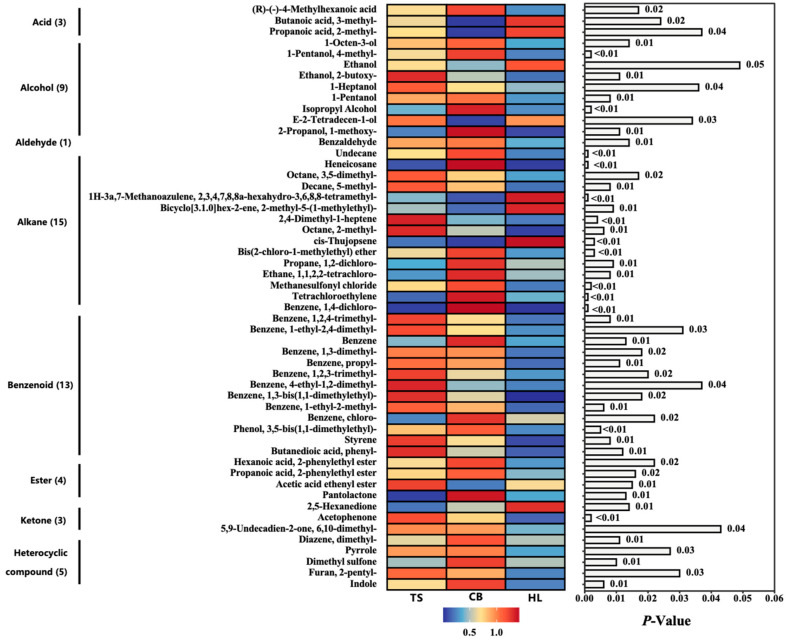
Volatile flavor compound composition in whole eggs from Hy-line Brown, Taihe, and Crossbred black-boned silky fowl. TS, Taihe black-boned silky fowl; CB, crossbred black-boned silky fowl; HL, Hy-line Brown.

**Table 1 foods-13-03308-t001:** Physical properties of Hy-line Brown, Taihe, and crossbred black-boned silky fowl eggs.

Egg Characteristics	TS		CB		HL		*p*
	Mean	Range	Mean	Range	Mean	Range	
Weight (g)	36.81 ± 0.39 ^C^	30.6~43.3	45.37 ± 0.52 ^B^	38~54.5	57.43 ± 0.47 ^A^	50.8~69.6	<0.01
Albumen height (mm)	3.54 ± 0.18 ^C^	2.5~9.0	3.96 ± 0.08 ^B^	2.5~5.3	7.74 ± 0.15 ^A^	3.3~9.8	<0.01
Haugh units	66.07 ± 1.29 ^B^	53.2~100.7	65.63 ± 0.88 ^B^	46.3~74.6	88.39 ± 0.98 ^A^	54.8~100	<0.01
Yolk color	11.65 ± 0.13 ^A^	10~14	7.53 ± 0.10 ^B^	6~9	5.92 ± 0.08 ^C^	4~7	<0.01
Yolk (%)	33.68 ± 0.45 ^A^	31.6~35.63	34.53 ± 0.61 ^A^	32.69~38.2	23.21 ± 0.31 ^B^	22.25~24.3	<0.01
Shell thickness (mm)	0.35 ± 0.00 ^A^	0.28~0.42	0.28 ± 0.01 ^B^	0.19~0.42	0.34 ± 0.00 ^A^	0.26~0.38	<0.01
Shell strength (kgf)	3.48 ± 0.08 ^B^	1.82~5	3.25 ± 0.12 ^B^	0.9~5.21	5.36 ± 0.09 ^A^	3.09~6.67	<0.01

All results are presented as mean values (*n* = 60) ± standard error. TS, Taihe black-boned silky fowl; CB, crossbred black-boned silky fowl; HL, Hy-line Brown. ^A–C^ Means in the same line with different superscripts were significant at *p* < 0.01.

**Table 2 foods-13-03308-t002:** Amino acid composition of egg whites from Hy-line Brown, Taihe, and crossbred black-boned silky fowl (mg/g).

Index	TS		CB		HL		*p*
	Mean	Range	Mean	Range	Mean	Range	
Lysine	5.82 ± 0.13 ^A^	5.45~6.39	3.39 ± 1.12 ^B^	0.23~8.48	0.46 ± 0.10 ^C^	0.25~0.98	<0.01
Tryptophan	0.09 ± 0.01 ^A^	0.06~0.14	0.06 ± 0.01 ^B^	0.02~0.11	0.02 ± 0.00 ^B^	0.02~0.03	<0.01
Phenylalanine	10.78 ± 0.94 ^A^	7.15~15.32	7.09 ± 1.06 ^B^	3.86~12.14	4.07 ± 0.35 ^C^	3.16~5.83	<0.01
Methionine	1.06 ± 0.26	0.04~2	1.25 ± 0.19	0.73~2.36	0.80 ± 0.07	0.63~1.15	0.29
Threonine	10.64 ± 0.66 ^A^	8.16~13.69	6.17 ± 1.25 ^B^	1.95~10.55	2.45 ± 0.38 ^C^	1.44~4.52	<0.01
Isoleucine	3.11 ± 0.23 ^A^	2.33~4.22	2.68 ± 0.42 ^A^	1.37~5.09	1.62 ± 0.12 ^B^	1.29~2.11	<0.01
Leucine	0.96 ± 0.05 ^B^	0.82~1.27	1.18 ± 0.13 ^B^	0.77~1.65	1.56 ± 0.06 ^A^	1.35~1.79	<0.01
Valine	8.53 ± 0.42 ^A^	6.89~10.39	5.66 ± 0.90 ^B^	2.64~8.9	2.89 ± 0.33 ^C^	1.97~4.58	<0.01
Alanine	1.68 ± 0.07 ^A^	1.4~1.95	0.83 ± 0.23 ^B^	0.21~1.69	0.21 ± 0.03 ^C^	0.14~0.35	<0.01
Asparagine	0.17 ± 0.01 ^A^	0.13~0.24	0.08 ± 0.03 ^B^	0.01~0.17	0.01 ± 0.00 ^C^	0.005~0.02	<0.01
Aspartic acid	36.16 ± 2.18 ^A^	28.05~47.19	17.54 ± 5.87 ^B^	1.38~35.69	1.62 ± 0.28 ^C^	0.86~3.17	<0.01
Cysteine	0.0024 ± 0.00 ^A^	0.002~0.003	0.0023 ± 0.00 AB	0.002~0.002	0.0022 ± 0.00 ^B^	0.002~0.002	<0.01
Glutamic acid	12.68 ± 0.75 ^A^	9.83~15.91	8.47 ± 1.56 ^B^	3.5~14.42	3.82 ± 0.40 ^C^	2.9~6.07	<0.01
Glutamine	0.06 ± 0.01 ^A^	0.05~0.07	0.05 ± 0.02 ^A^	0.01~0.13	0.01 ± 0.00 ^B^	0.01~0.02	<0.01
Glycine	13.49 ± 1.08 ^A^	9.72~19.68	5.84 ± 1.95 ^B^	0.47~14.08	0.63 ± 0.13 ^C^	0.3~1.38	<0.01
Proline	5.43 ± 0.30 ^A^	4.45~6.92	3.47 ± 0.61 ^B^	1.4~5.73	1.58 ± 0.18 ^C^	1.04~2.44	<0.01
Serine	15.57 ± 0.72 ^A^	12.62~18.46	8.56 ± 1.99 ^B^	2.44~15.55	2.68 ± 0.37 ^C^	1.67~4.63	<0.01
Histidine	20.46 ± 0.51 ^A^	18.89~23.43	11.07 ± 3.06 ^B^	1.87~22.82	2.34 ± 0.51 ^C^	1.14~5.41	<0.01
Arginine	9.80 ± 0.13 ^A^	9.1~10.21	6.05 ± 1.19 ^B^	2.26~10.7	2.40 ± 0.39 ^C^	1.39~4.75	<0.01
Tyrosine kinase	11.30 ± 1.11 ^A^	8.1~17.56	6.65 ± 1.17 ^B^	2.7~12	3.10 ± 0.39 ^C^	2.16~5.16	<0.01

All results are presented as mean values (*n* = 8) ± standard error. TS, Taihe black-boned silky fowl; CB, crossbred black-boned silky fowl; HL, Hy-line Brown. ^A–C^ Means in the same line with different superscripts were significant at *p* < 0.01.

**Table 3 foods-13-03308-t003:** Nutritive evaluation indices for protein and amino acids in egg whites from different breeds.

Index	TS		CB		HL	
	RAA	RCAA	RAA	RCAA	RAA	RCAA
Leucine	0.08	0.08	0.16	0.16	0.41	0.43
Valine	1.12	1.15	1.14	1.17	1.17	1.22
Methionine + Cysteine	0.24	0.25	0.44	0.45	0.57	0.59
Isoleucine	0.54	0.56	0.72	0.74	0.87	0.91
Threonine	2.23	2.30	1.98	2.05	1.59	1.66
Phenylalanine + Tyrosine	2.83	2.91	2.70	2.78	2.83	2.96
Lysine	0.64	0.65	0.57	0.59	0.15	0.16
Tryptophan	0.07	0.07	0.08	0.07	0.06	0.06
EAAI	49.64		57.19		56.04	
EAA/TAA	24.43		33.12		43.77	
EAA/NEAA	32.37		51.66		78.20	

TS, Taihe black-boned silky fowl; CB, crossbred black-boned silky fowl; HL, Hy-line Brown. RAA, ratio of amino acid; RCAA, relative coefficient of amino acid; EAA, essential amino acid; EAAI, essential amino acid index; TAA, total amino acid; NEAA, nonessential amino acid.

**Table 4 foods-13-03308-t004:** Lipid composition in egg yolks from Hy-line Brown, Taihe, and crossbred black-boned silky fowl.

Index	TS		CB		HL		*p*
	Mean	Range	Mean	Range	Mean	Range	
Crude fat (mg/g)	340.69 ± 16.12	274.6~396.2	334.93 ± 20.36	289.2~473.1	334.87 ± 11.16	297.3~375.4	0.96
Triglyceride (mg/g)	14.93 ± 0.44	13.24~16.65	14.91 ± 0.49	13.15~17.63	15.92 ± 0.94	11.12~19.48	0.48
Total Cholesterol (mg/g)	8.34 ± 0.12	7.79~8.87	8.05 ± 0.73	6.32~12.8	7.14 ± 1.17	5.42~9.02	0.22
PC (mg/g)	1.60 ± 0.07 ^A^	1.34~1.86	1.16 ± 0.03 ^B^	1.01~1.28	1.03 ± 0.02 ^B^	0.97~1.12	<0.01
PE (mg/g)	11.28 ± 0.31 ^A^	9.51~12.23	10.63 ± 0.28 ^A^	9.19~11.42	8.92 ± 0.26 ^B^	7.60~9.87	<0.01
SM (mg/g)	1.52 ± 0.04 ^A^	1.27~1.65	1.18 ± 0.03 ^B^	1.07~1.31	1.06 ± 0.03 ^B^	0.88~1.11	<0.01
Cer (mg/g)	1.35 ± 0.19 ^A^	0.69~2.13	0.83 ± 0.10 ^B^	0.49~1.25	0.62 ± 0.04 ^B^	0.46~0.78	<0.01
25-OH-VD2 ^a^ (ng/g)	29.25 ± 0.32	28.33~31.3	29.20 ± 0.34	28.21~31.02	28.61 ± 0.08	28.25~28.93	0.21
25-OH-VD3 ^a^ (ng/g)	21.60 ± 2.06 ^A^	14.09~28.5	23.51 ± 1.61 ^A^	15.66~28.29	15.18 ± 1.52 ^B^	10.06~23.42	<0.01
VK ^a^ (ng/g)	2.77 ± 0.07	2.45~3.03	2.64 ± 0.10	2.3~3.1	2.70 ± 0.08	2.36~3	0.55
VE ^a^ (μg/g)	2.68 ± 0.19 ^B^	1.82~3.43	6.61 ± 1.06 ^A^	2.8~10.69	3.33 ± 0.90 ^B^	1.76~9.55	<0.01
VA ^a^ (μg/g)	3.15 ± 0.19	2.50~4.28	3.60 ± 0.21	2.90~4.49	3.30 ± 0.27	2.55~4.81	0.36

All results are presented as mean values (*n* = 8) ± standard error. TS, Taihe black-boned silky fowl; CB, crossbred black-boned silky fowl; HL, Hy-line Brown. PC, phosphatidylcholine; PE, phosphatidylethanolamine; Cer, ceramide; SM, sphingomyelin. ^A,B^ Means in the same line with different superscripts were significant at *p* < 0.01. ^a^ Represents a fat-soluble vitamin, including 25-OH-VD2, 25-OH-VD3, vitamin K (VK), vitamin C (VC), and vitamin A (VA).

**Table 5 foods-13-03308-t005:** Nutritional analysis of fatty acid data obtained from egg yolk samples (mg/g).

Index	TS		CB		HL		*p*
	Mean	Range	Mean	Range	Mean	Range	
∑FA	205.32 ± 3.86	190.9~226.8	195.97 ± 11.43	146.4~232.4	209.35 ± 9.56	158.1~239.2	0.56
∑SFA	76.45 ± 1.70	71.97~85.61	69.82 ± 4.17	51.50~83.49	75.49 ± 3.42	56.92~86.78	0.32
∑MUFA	86.96 ± 1.48	84.02~97.58	83.32 ± 4.36	65.69~99.09	95.0.4 ± 4.49	71.1~109.43	0.11
∑PUFA	39.17 ± 0.95	34.87~43.65	42.83 ± 2.99	29.18~54.04	38.82 ± 1.89	30.10~44.70	0.35
∑ω3-PUFA	1.82 ± 0.05 ^A^	1.59~2.02	1.83 ± 0.16 ^A^	1.29~2.31	1.24 ± 0.05 ^B^	1.00~1.40	<0.01
∑ω6-PUFA	37.36 ± 0.9	33.28~41.63	41.00 ± 2.87	27.89~51.73	37.58 ± 1.84	29.1~43.3	0.38
ω3/ω6	0.049 ± 0.00 ^A^	0.04~0.05	0.045 ± 0.00 ^B^	0.04~0.05	0.033 ± 0.00 ^C^	0.03~0.04	<0.01
C18:3 N3	1.05 ± 0.04 ^A^	0.89~1.21	1.07 ± 0.07 ^A^	0.76~1.38	0.70 ± 0.03 ^B^	0.56~0.80	<0.01
EPA	0.037 ± 0.00 ^A^	0.03~0.05	0.030 ± 0.00 ^B^	0.02~0.04	0.025 ± 0.00 ^B^	0.02~0.03	<0.01
DHA	0.45 ± 0.01 ^A^	0.42~0.51	0.45 ± 0.03 ^A^	0.31~0.55	0.30 ± 0.01 ^B^	0.24~0.33	<0.01
AI	0.45		0.43		0.45		

All results are presented as mean values (*n* = 8) ± standard error. TS, Taihe black-boned silky fowl; CB, crossbred black-boned silky fowl; HL, Hy-line Brown; ∑FA: total fatty acids; ∑SFA: total saturated fatty acids; ∑MUFA: total monounsaturated fatty acids; ∑PUFA: total polyunsaturated fatty acids; ω-3 PUFA, ω-3 polyunsaturated fatty acids; ω-6 PUFA, ω-6 polyunsaturated fatty acids; C18:3N3, α-linolenic acid; EPA, eicosapentaenoic acid; DHA, docosahexaenoic acid; AI, atherogenic index. ^A–C^ Means in the same line with different superscripts were significant at *p* < 0.01.

**Table 6 foods-13-03308-t006:** Mineral element composition in whole eggs from Hy-line Brown, Taihe, and Crossbred black-boned silky fowl (μg/g).

Index	TS		CB		HL		*p*
	Mean	Range	Mean	Range	Mean	Range	
Mg	108.32 ± 1.65 ^B^	99.7~116.1	103.89 ± 1.01 ^B^	99.8~107.2	118.86 ± 2.47 ^A^	110.1~128.2	<0.01
Ca	611.75 ± 28.47 ^A^	449.4~726.8	609.94 ± 15.26 ^A^	527.1~665.4	491.55 ± 6.09 ^B^	453.8~509.4	<0.01
P	1206.20 ± 219.60	730~2548	975.45 ± 153.36	562~1709	1069.85 ± 56.29	786~1262	0.59
Mn	0.43 ± 0.03 ^B^	0.24~0.55	0.52 ± 0.02 ^A^	0.43~0.62	0.30 ± 0.01 ^C^	0.24~0.35	<0.01
Fe	5.23 ± 1.11 ^B^	0.55~8.82	11.44 ± 0.51 ^A^	9.77~13.71	9.10 ± 0.50 ^A^	7.1~11.12	<0.01
Cu	0.73 ± 0.03 ^A^	0.55~0.84	0.67 ± 0.02 ^A^	0.58~0.72	0.58 ± 0.01 ^B^	0.53~0.60	<0.01
Zn	13.93 ± 0.63 ^A^	10.22~15.93	13.68 ± 0.35 ^A^	12.25~14.97	9.75 ± 0.25 ^B^	8.73~10.45	<0.01
Se	0.19 ± 0.02 ^A^	0.12~0.27	0.11 ± 0.01 ^B^	0.06~0.15	0.06 ± 0.01 ^C^	0.03~0.10	<0.01
Cr	0.008 ± 0.00	0.0001~0.004	0.016 ± 0.00	0.004~0.035	0.019 ± 0.00	0.002~0.035	0.052
Mo	0.048 ± 0.00 ^B^	0.03~0.06	0.049 ± 0.00 ^B^	0.04~0.06	0.11 ± 0.01 ^A^	0.07~0.18	<0.01
Pb	0.001 ± 0.00	ND (4)~0.001	0.002 ± 0.00	ND (2)~0.004	0.003 ± 0.00	ND (4)~0.006	0.29

All results are presented as mean values (*n* = 8) ± standard error. TS, Taihe black-boned silky fowl; CB, crossbred black-boned silky fowl; HL, Hy-line Brown; ND, not detectable. ^A–C^ Means in the same line with different superscripts were significant at *p* < 0.01.

**Table 7 foods-13-03308-t007:** Volatile flavor compounds composition in whole eggs from Hy-line Brown, Taihe, and crossbred black-boned silky fowl (%).

Index	TS		CB		HL		*p*
	Median	IQR	Median	IQR	Median	IQR	
Acid	1.7034 ^a^	9.2122–0.7379	0.5212 ^b^	0.7451–0.3794	10.0057 ^c^	19.5243–1.315	0.03
Alcohol	30.657	48.551–17.674	28.608	32.876–24.867	49.6915	52.147–45.491	0.06
Aldehyde	0.637 ^a^	0.8905–0.3159	0.7078 ^a^	0.8769–0.4385	0.1505 ^b^	0.3070–0.1188	0.02
Alkane	20.4833 ^ab^	29.433–13.115	30.4839 ^a^	32.811–23.492	5.7166 ^b^	14.260–3.719	0.01
Benzenoid	13.1383 ^A^	24.875–4.2683	9.8257 ^A^	10.617–7.94	0.9951 ^B^	3.5877–0.5387	<0.01
Ester	4.447 ^a^	7.955–2.6417	1.0528 ^b^	1.2773–0.9173	2.3751 ^a^	3.9776–1.8472	0.01
Ketone	3.748	5.0045–2.6087	4.8463	5.8476–3.5094	6.9086	10.119–4.2931	0.15
Heterocyclic compound	8.5117 ^b^	11.479–5.9905	22.767 ^a^	24.758–18.14	7.726 ^b^	17.86~6.2544	0.02

All results are presented as median values (*n* = 8) and IQR. TS, Taihe black-boned silky fowl; CB, crossbred black-boned silky fowl; HL, Hy-line Brown; IQR, interquartile range. ^A,B^ Means in the same line with different superscripts were significant at *p* < 0.01. ^a–c^ Means in the same line with different superscripts were significantly different (*p* < 0.05).

**Table 8 foods-13-03308-t008:** ROAV analysis of volatile flavor compounds in whole eggs from Hy-line Brown, Taihe, and Crossbred black-boned silky fowl.

Index	ROAV			Description of Flavor
	TS	CB	HL	
Ethyl isovalerate	100	15.88	100	Fruity (apple, pineapple, banana)
1-Octen-3-ol	83.82	100	5.90	Fishy, earthy, grassy
Ethyl 2-methylpropionate	6.42	2.66	5.49	Fruity (osmanthus, apple, peach)
1-Hexanol	10.89	5.08	1.35	Herbal taste, grassy
1-Heptanol	1.94	1.09	/	Natural smell, nutty
2-Pentylfuran	2.96	2.16	/	Phase bean scent
Styrene	1.10	/	/	Balsam, floral
1,2-Dichloropropane	/	1.20	/	Sweet smell
1,4-Dichlorobenzene	/	1.85	/	Aromatic
Indole	/	1.29	/	Mothballs, dung

TS, Taihe black-boned silky fowl; CB, crossbred black-boned silky fowl; HL, Hy-line Brown; ROAV, relative odor activity value.

## Data Availability

The data presented in this study are available on request from the corresponding author. The data are not publicly available due to privacy restrictions.

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
