# Peer review of "Characterization and Evaluation of Taihe Black-Boned Silky Fowl Eggs Based on Physical Properties, Nutritive Values, and Flavor Profiles"

_foods, 2024, doi:10.3390/foods13203308_

Round 1

Reviewer 1 Report

Comments and Suggestions for Authors

Remarks

Introduction: 

Line 61: Please explain, what is understood as eggshell quality.

Materials and methods:

Subsection 2.2:

Please provide more details. How are defined color units? How is defined eggshell strength.

Subsection 2.3.3:

Please specify GC-MS/MS equipment, software and protocol.

Subsection 2.3.4.

Please specify LC-MS/MS equipment, software and protocol.

Subsection 2.4.3:

Please express INQ calculation as equation instead of text fragment.

Subsection 2.4.4:

Please express TAV calculation as equation instead of text fragment.

Please provide websites of databases and cite articles describing them (if available). 

Results:

Table 1. Please provide units of all parameters characterizing eggs.

Word "extremely" about ststistical significant is not necessary. Please write simply "significant at P < 0.01". The same remark concerns all tables.

Figures  1 nad 2. Are differences statistically significant? If yes please indicate it. If not please state it.

Heat maps can be enlarged. Please indicate program used for construction of heat maps.

Please add list of all abbreviations at the end of text (before references). it is often practiced in MDPI journals.

Reviewer 2 Report

Comments and Suggestions for Authors

Revisions and comments

Dear Authors,

I commend your efforts in characterizing the physical, nutritional, and flavor properties of Taihe black-boned silky fowl eggs. Overall, I find the manuscript to be clear, the results are interesting, and the discussion is thorough and appropriate.

I only have some minor comments:

1) I believe that declaring the statistical analysis was performed on Windows is unnecessary (line 246).

2) Did you perform a normality test to analyze the data distribution in the study? If so, please specify which test was used and include this information in the manuscript.

3) Did you perform an analysis to assess homoscedasticity? If so, please specify which test was used and include this information in the manuscript.

4) The authors mentioned that they performed a Kruskal-Wallis test, yet all results are presented as mean ± standard error. When reporting non-parametric data, the median and interquartile range are more appropriate descriptors. Please revise.

5) Please specify the post hoc test employed after the Kruskal-Wallis test.

6) In relation to my previous comment, the authors state that data derived from volatile flavor compounds were analyzed using a Kruskal-Wallis test. However, in Table 7, the data are presented as mean ± standard error. Please revise this to reflect the appropriate statistical descriptors.

7) The terms "Mean" and "Average" are used inconsistently throughout the manuscript. Please homogenize the terminology.

8) Figures 4, 5, and 7 should be enlarged for better readability. In their current size, it is difficult to adequately interpret the figures.

9) Is there truly no statistical difference between the acid index of TS (6.81 ± 3.65ab) and CB (0.67 ± 0.16b)? Please review this and change if necessary.

10) It is recommended to include a section at the end of the discussion that addresses the main strengths and limitations of the study. This would help to highlight the key findings and suggest future research directions on this topic.

Reviewer 3 Report

Comments and Suggestions for Authors

The manuscript titled "Characterization and evaluation of Taihe black-boned silky fowl eggs based on physical properties, nutritive values, and flavor profiles" presents a comprehensive evaluation of eggs from Taihe black-boned silky fowl (TS) and compares them to crossbred black-boned silky fowl (CB) and commercial Hy-line Brown (HL) eggs. The study explores these eggs' physical, nutritional, and flavor properties, analyzing parameters such as amino acids, fatty acids, minerals, and volatile flavor compounds. The results suggest that despite their smaller size, TS eggs have unique nutritional and flavor characteristics that may offer health benefits, such as being a good source of protein for young children and beneficial lipids for cardiovascular health. The study offers a novel contribution by characterizing the eggs of Taihe black-boned silky fowl and comparing them with commercial breeds. While many studies have focused on commercial egg production, this research highlights a lesser-known breed with potentially significant health benefits, which is relatively innovative. However, the study would benefit from a deeper exploration of how these findings could influence consumer behavior and industry practices. More comments are presented below:

·        The manuscript could benefit from additional information on the selection criteria for the eggs used in the analysis. Were the hens all from similar age groups, or were there other factors considered (e.g., diet, environment)?

·        The data on amino acids is extensive, but it would be useful to relate these findings more clearly to potential health implications for different age groups. For example, how does the amino acid composition specifically benefit the recommended demographic groups (children vs. adults)?

·        Could you provide more detail on the potential mechanisms that link the unique flavor profiles of TS eggs to their nutritional composition? For instance, how do specific amino acids or fatty acids contribute to the distinct taste of TS eggs?

·        You mention that the TS eggs could serve as a beneficial dietary lipid source. Have any clinical or consumer studies support these claims, particularly in relation to cardiovascular health?

·        The study suggests that TS eggs are of lower quality in terms of Haugh units and shell strength. What steps could be taken in breeding or feed formulation to improve these aspects without losing the other benefits?

Comments on the Quality of English Language

The English is understandable.
